# Industrial Scale Gene Editing in *Brassica napus*

Andrew Walker [1], Javier Narváez-Vásquez [2], Jerry Mozoruk [1], Zhixia Niu [1], Peter Luginbühl [1], Steve Sanders [1], Christian Schöpke [1], Noel Sauer [1,*], Jim Radtke [1], Greg Gocal [1] and Peter Beetham [1]

1  Cibus Inc., San Diego, CA 92121, USA; awalker@cibus.com (A.W.); jmozoruk@cibus.com (J.M.); zniu@cibus.com (Z.N.); pluginbuhl@cibus.com (P.L.); ssanders@cibus.com (S.S.); cschopke@cibus.com (C.S.); jradtke@cibus.com (J.R.); ggocal@cibus.com (G.G.); pbeetham@cibus.com (P.B.)
2  Botany and Plant Sciences Department, University of California, Riverside, CA 92521, USA; jnarvaez@ucr.edu
*  Correspondence: nsauer@cibus.com

**Abstract:** In plants, an increasing number of traits and new characteristics are being developed using gene editing. Simple traits represented by a single gene can be managed through backcross breeding, but this is typically not the case for more complex traits which may result from the function of a large number of genes. Here, we demonstrate two case studies of improving oleic oil content and developing pod shatter reduction in *Brassica napus* by using gene editing tools on an industrial scale. There are four *BnaFAD2* genes involved in oleic oil content and eight *BnaSHP* genes involved in pod shatter tolerance. In order to develop these two traits, we delivered nuclease ribonucleoproteins with Gene Repair OligoNucleotides (GRONs) into protoplasts, with subsequent regeneration into plants on an industrial scale, which encompassed robust tissue culture protocols, efficient gene editing, robotics sampling, and molecular screening, vigorous plant regeneration, growth, and phenotyping. We can produce precise loss-of-function-edited plants with two improved agronomically important complex traits, high oleic oil or pod shatter reduction, in elite canola varieties within 1–3 years, depending on the trait complexity. In the edited plants carrying loss of function of four *BnaFAD2* genes, the seed fatty acid oleic acid content reached 89% compared to 61% in the non-edited wildtype control. The plants carrying eight edited *BnaSHP* genes achieved 51% pod shatter reduction in multiple year field testing in the target environment compared to the wildtype control.

**Keywords:** canola; *Brassica napus*; gene editing; multiplex gene editing; non-transgenic gene editing; pod shatter; high oleic oil

## 1. Introduction

For more than two decades, gene editing has been a developing field. Gene targeting, as it was once known, included techniques such as chimeraplasty and oligo-directed mutagenesis (ODM; [1]). Around the same time that these techniques were developed, rare cutting endonucleases known as meganucleases were deployed, followed by engineered zinc finger nucleases, transcription activator-like effector nucleases (TALENs), and most recently, clustered regulatory interspaced short palindromic repeat (CRISPR)-associated protein (Cas) (CRISPR/Cas), which has become synonymous with the field of gene editing [2,3]. Extending these gene editing tools, additional nucleases such as Cas12a (Cpf1), with its differing PAM (protospacer adjacent motif) requirement compared to Cas9, have further expanded the available target sites for gene editing, and new approaches, such as covalently linked activators, repressors, methylases, and base editors, as well as prime editing, have broadened the options to precisely fine-tune gene sequences, including gene expression [3–5].

As with all gene editing methods involving DNA strand breaks, two different types of DNA repair mechanisms, non-homologous end-joining (NHEJ) and homology-directed repair (HDR), are exploited. In plant cells, NHEJ is the favored repair pathway, with its imprecise repair mechanism frequently making small insertions or deletions (indels) and to

a lesser degree, nucleotide substitutions [6–15]. Indels, if located within a coding sequence, often result in a frame-shift mutation that impairs gene function.

While to date, most gene editing applications in canola have exploited the NHEJ pathway [16], specific point mutations can be achieved in a target gene using the less preferred HDR pathway. This method includes both the delivery of gene editing reagents to make a double strand break as well as a repair template to guide the gene editing outcome [17].

Gene editing of more complex traits may utilize this less active HDR pathway by employing a suite of advanced gene editing and cell biology techniques. Chemically synthesized DNA repair templates, also known as Gene Repair OligoNucleotides (GRONs), along with double-stranded DNA cleaving nuclease proteins such as CRISPR/Cas9 can mediate one or a few very specific nucleotide changes in the target DNA sequences that may be needed to develop more complex traits [17]. These DNA repair templates are comprised of DNA bases, as well as chemical moieties, which are designed to cause mismatched base pair(s) at the targeted DNA location and to prevent incorporation, respectively. The mismatched base pair, along with the double-stranded break in the target DNA, acts as a signal to attract the cell's repair system to that site and correct the designated nucleotide(s) within the DNA. After the correction process is complete, the gene editing reagents are degraded through natural processes within the plant cell. Once a plant cell, such as a protoplast, is edited, it can be cultured to develop into a microcallus (a small clump of undifferentiated cells), a visible callus, and eventually individual shoots. Shoots harboring the stably inherited, intended changes in their DNA can be evaluated for the desired phenotype both in the greenhouse and in the field.

To date, most applications of gene editing in canola have involved the delivery of DNA-based CRISPR/Cas9 gene editing reagents which are first integrated into the genome and expressed as a transgenic construct, and then segregated away by breeding as null segregants to leave only the desired loss-of-function (LOF) allele(s) [16]. However, this may be difficult to achieve in circumstances in which (1) DNA-based gene editing reagents may incorporate within the cut site or in a region tightly linked to the target allele or (2) in the case of complex multiplex/multitarget traits, large populations of at least 4n (where n is the number of gene-edited loci) are required to obtain null segregants with these traits. Further, while DNA-based gene editing reagents allow the ability to enrich for transformants using selective agents such as kanamycin, hygromycin, and BASTA®, the time in culture required for efficient selection feeds a progressive gene editing process that can lead to chimeras. For instance, Yang et al. [18] observed that more than 30% of the transgenic T0 plants analyzed were chimeric. This progressive gene editing effect may occur at a low frequency during the growth of the transgenic plants and even in subsequent generations where the remaining wildtype (WT) alleles can be gene edited [19].

An alternative delivery method is the direct delivery of DNA-free reagents such as CRISPR-Cas9 ribonucleoproteins (RNP) or mRNA-based Cas9. The methods most amenable to this DNA-free approach include biolistic (particle bombardment) and direct delivery of gene editing reagents into protoplasts via polyethylene glycol (PEG), electroporation, or by other means. All of these DNA-free delivery approaches avoid the use of Agrobacterium, the method traditionally used for the majority of crop species [20].

Protoplasts represent a source material with several benefits for DNA-free gene editing. For example, protoplasts can be isolated and have gene editing reagents delivered into them in just a few days, allowing for rapid evaluation of target site mutation efficiency. Protoplasts allow for the direct delivery of in vitro assembled active RNPs, bypassing the use of exogenous DNA for the delivery of gene editing reagents [21–23]. Further, protoplasts can be used to generate thousands of independent events and do not require a selectable marker, thus enabling plants to be regenerated without a transgenic step (without incorporation of any foreign DNA [24,25]). Thus, by sourcing protoplasts as a starting material, delivery efficiencies can be high, and experimental timelines can be drastically shortened [26–29]. Further, robotic platforms have recently been developed for protoplast

isolation and delivery, which opens the possibility of automated high-throughput gene editing capabilities [27].

One disadvantage of a protoplast-based gene editing platform is that it can be challenging to regenerate plants, particularly for elite varieties of many commercially relevant species. Despite these limitations, protoplasts from at least six crops (flax, rice, wheat, maize, lettuce, and tomato), in addition to *Arabidopsis* and tobacco, have been used to evaluate gene editing reagents using CRISPR/Cas9-based systems [10,17,21,30,31]. Successful induction of CRISPR-Cas9 mutations by DNA-free CRISPR-Cas9 RNPs has been demonstrated for *Arabidopsis*, tobacco (*Nicotiana tabacum*), rice (*Oryza sativa*), lettuce (*Lactuca sativa*), and maize (*Zea mays*) [21,32]. More recently, CRISPR/Cas12a and base editing systems delivered to protoplasts as pre-assembled RNPs have also been used to achieve targeted mutagenesis and subsequent whole plant regeneration [21,22,33,34].

Protoplast-based methods have similarly been established for canola. Kartha et al. [35] and Gocal et al. [36] described a canola protoplast isolation protocol where plantlets could be regenerated from protoplasts, and work by Wang et al. [37] also demonstrated a transient canola protoplast transfection system. Lin et al. [6] further showed that canola can be effectively mutated using protoplasts from 2- to 3-week-old cotyledons of in vitro grown seedlings.

As gene editing technologies continue to evolve, and the efficiency of plant regeneration in cell culture continues to improve, protoplasts will continue to be a viable source material for the delivery of gene editing reagents for trait development.

Here, we introduce the concept of industrialized gene editing (IGE), which may use technologies that precisely and efficiently produce non-transgenic traits in plants by way of an efficient protoplast-based platform. IGE evolved through significant advancements in both gene editing and tissue culture technologies (Figure 1). It is a suite of technologies representing the ability to isolate single plant cells (protoplasts), make the desired genetic edits in those cells, and regenerate those edited cells into plants. With this, IGE can establish a flexible, scalable, and reproducible process that can be used to develop customized plant products with multiple desirable traits to meet the needs across the agricultural value chain. The potential of IGE lies in its ability to develop complex traits involving multiplex edits (i.e., multiple edits in a gene target and/or edits in multiple gene targets) which, for example, address diseases and pests, enhance crop yields for growers, improve the productivity of processors, produce healthier and more nutritious foods for consumers, and help in the battle against climate change.

Herein, we demonstrate the use of a single-cell approach in IGE which includes isolating protoplasts from elite varieties of canola and delivering GRONs (chemically protected DNA templates) and nuclease reagents into these single cells to orchestrate one or many targeted spelling changes which result in the simultaneous editing of four to eight LOF alleles for two important commercially relevant traits, high oleic oil and pod shatter reduction. Plants regenerated from these gene-edited protoplasts are examined for the desired phenotype in both the greenhouse and field settings.

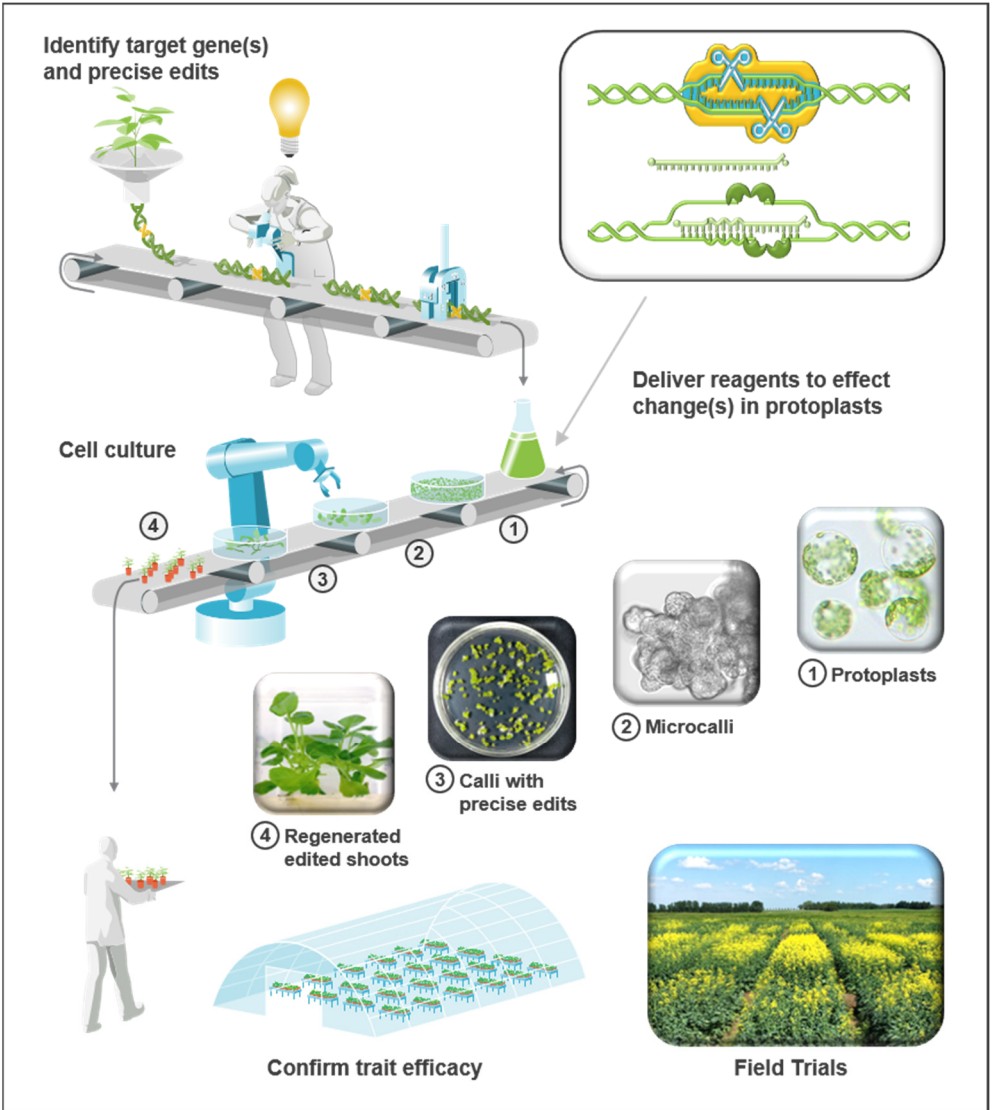

**Figure 1.** The concept of industrialized gene editing (IGE) encompasses the steps for the application of advanced large-scale gene editing and cell biology techniques. In the case of canola (*B. napus*), this includes the identification of target genes whose gene editing is expected to lead to improved traits, the assembly of the gene editing reagents (e.g., GRONs and nucleases) to achieve the desired edit(s), the delivery of reagents to protoplasts, the identification of gene-edited calli or shoots, the confirmation of trait efficacy in the greenhouse, and finally, field trials to validate the new trait. The cartoon represents in schematic form these different steps. The basis for IGE in canola is a robust protoplast-to-plant regeneration system (see steps 1–4) and the inclusion of automated processes which enable high-throughput tissue culture. The cartoon images are for illustrative purposes only and the layout may not be an exact representation.

## 2. Materials and Methods

### 2.1. Generation of Gene-Edited Plants

The editing reagents used to develop LOF mutations in the *BnaFAD2* and *BnaSHP* genes consisted of Cas9 protein complexed with a sgRNA, forming a ribonucleoprotein (RNP), along with GRONs which contain precise gene specific mutations ($N_{+1}$ insertion, $N_{-1}$, and $N_{-2}$ deletions; Table S1). Cas9 protein and GRONs were obtained from Integrated DNA Technologies (IDT; Coralville, IA, USA). sgRNA was either produced internally using the HiScribe T7 High Yield RNA Synthesis Kit (NEB; Ipswich, MA, USA) or obtained commercially from IDT. RNPs and GRONs were introduced into the protoplasts by PEG-

mediated delivery, following the methodologies described in [17]. Canola protoplasts were isolated from the leaves of micropropagated haploid plants following a standard protocol [38]. Following the delivery of gene editing reagents, the protoplasts were cultured in liquid medium ($1.25 \times 10^5$ cells/mL) and incubated in the dark at 25 °C. At 6–8 weeks, protoplast-derived microcalli were transferred to solidified shoot regeneration medium and incubated under a 16 h photoperiod (90–110 $\mu$mol/m$^2$/s). Shoots started developing after 2–4 weeks. Genomic DNA isolated from leaf samples of fully differentiated shoots was analyzed by targeted amplicon sequencing to determine the presence of intended mutations in each of the targeted genes. Shoots with targeted mutations in individual and multiple genes were screened for off-target mutations. After flow cytometric determination of ploidy, dihaploids were micropropagated, followed by transfer to soil in a growth chamber. Hardened plants were transferred to the greenhouse and grown to maturity (seed setting).

### 2.2. Genotype Identification by NGS

Genomic DNA was extracted from regenerated shoots using the PureGenome Plant genomic DNA kit as per the manufacturer's recommendations (Aline Biosciences, Woburn, MA, USA). Amplicons were generated with primers flanking the intended edit position using Phusion$^\circledR$ polymerase (New England Biolabs, Ipswich, MA, USA) and 50 ng of genomic DNA. The amplicons were then subjected to deep sequence genotyping using a $2 \times 250$ bp Illumina platform (Illumina, San Diego, CA, USA). Genotype identification analysis of screened shoots was performed using CLC Genomics Workbench (CLC Bio, Boston, MA, USA) and custom workflows to identify precise gene edits.

### 2.3. Off-Target Analysis

Computationally predicted reagent-mediated off-target loci for both *BnaFAD2* and *BnaSHP* gRNAs were determined using Cas-OFFinder with the following parameters: NGG PAM, up to and including 3 mismatches and 1 mismatch with either an RNA or DNA bulge [39]. For the *BnaFAD2* genes, four predicted off-target loci were evaluated by whole genome sequencing (WGS). For the *BnaSHP* genes, 52 loci were evaluated by an amplicon deep sequencing approach. Mutations near the expected RNP cleavage site were considered off-target events.

### 2.4. Characterization of Plants with BnaFAD2 LOF Mutations

Plants with *BnaFAD2* LOF mutations that were devoid of off-target mutations at predicted loci and exhibited the desired ploidy were transferred to soil and hardened off in growth chambers prior to being transferred to a controlled greenhouse environment for seed increase and evaluation of the desired oil quality trait phenotype. While in the greenhouse, plants were assessed for overall true-to-type plant morphology. The vast majority of regenerated plants (more than 80–90%) derived from the gene editing process retained the same morphological characteristics as the parent line. Individual regenerated plants were genotyped to confirm the stable presence of the LOF allele(s) induced by the gene editing process. Genotyping results confirmed that target loci of selected individuals were homozygous for the LOF allele(s). Plants completed their growth cycle in the greenhouse and seeds were harvested from several gene-edited lines individually, representing multiple genotype combinations of the four genes targeted for the oil quality trait. Dried seeds of the greenhouse grown *BnaFAD2* gene-edited lines along with wildtype reference lines were evaluated for fatty acid composition using FAME (fatty acid methyl ester) analysis.

Cleaned canola seeds suspended in heptane were crushed with beads in a TissueLyser (Qiagen, Germantown, MD, USA) to extract the oil. Following centrifugation, the top organic phase was taken and saponified with sodium methoxide (0.5 N) to convert them into fatty acid methyl esters. After phase separation, the top layer was taken and diluted to be analyzed through gas chromatography flame ionization detection (GC-FID specifications: Inlet, 250 °C, 27 psi; total flow: 13.6 mL/min; split ratio: 10: 1; column type: capillary;

manufacturer: Phenomenex; model number: ZB FAME; column length: 20 m; diameter: 180 µM; film thickness: 0.15 µM; flow rate: 0.97 mL/min; FID temp: 250 °C).

### 2.5. Random Impact Test (RIT) for Pod Shatter

Pods were sampled from multiple plants per gene-edited line and analyzed in the lab, where they were subjected to a shaking force in the random impact test (RIT) using a Geno/Grinder (Spex, Metuchen, NJ, USA). Individual replicates were averaged and compared across genotypes. By comparing the amount of "breakage force" above the isogenic, wildtype background (% of additional BF; calculated by subtracting the amount of BF for the wildtype from the BF of the gene-edited line, then dividing by the BF of the wildtype and multiplying by 100), we were able to quantify the impact of the various combinations of mutant LOF alleles derived from our gene editing approach.

## 3. Results

### 3.1. Single Cell Approach—High Oleic Oil

Canola oil is high in unsaturated fatty acids comprised of monounsaturated oleic acid and an optimal 2:1 ratio of the polyunsaturated linoleic and linolenic acid, which is considered to be optimal for human consumption [40]. Canola oils with higher oleic acid (C18:1) contents are desirable because of their higher thermal stability, which can maintain preferred flavors in storage, extend shelf life, and enable new commercial and industrial applications. In plants, the stearoyl-acyl carrier protein desaturase encoded by the Fatty Acid Desaturase 2 (FAD2) genes catalyzes the desaturation of stearic acid (C18:0) to oleic acid (C18:1). The *B. napus* genome contains four FAD2 orthologues. Three copies appear intact and likely to be functionally redundant, while one copy, *BnaA01.FAD2b*, appears to be a non-functional pseudogene resulting from a single bp deletion at nucleotide 164 which leads to a premature stop codon at position 411 [41–45]. It was hypothesized that LOF mutations in the *BnaFAD2* genes could result in a high oleic oil phenotype.

Leveraging our protoplast regeneration system and multicomponent approach to gene editing, we sought to develop *BnaFAD2* LOF genotypes in canola using chemically modified single-stranded GRON templates containing either a $N_{+1}$ insertion, a $N_{-1}$ deletion, or a $N_{-2}$ deletion coupled with CRISPR/Cas9 RNP-based methodologies [46]. A gene editing event specified by these GRON templates would effectively make LOF mutations by altering the reading frame of the coding sequence.

To accomplish this, we co-delivered a template pool consisting of GRONs BnaFAD2-G01 through BnaFAD2-G12 (Table S1), along with CRISPR/Cas9 RNPs BnaFAD2-01 and BnaFAD2-02 (Table S2), into canola protoplasts using a PEG-mediated delivery approach [17]. Leaf samples from fully differentiated shoots derived from GRON plus RNP-treated protoplasts were analyzed by targeted amplicon deep sequencing to determine the occurrence of precise mutations in each of the four *FAD2* genes. Table 1 summarizes the number of shoots regenerated from two canola lines with each of the 15 possible LOF genotypes, including shoots with single and multiple edited *FAD2* genes. GRON-specified mutations ($N_{+1}$, $N_{-1}$, $N_{-2}$ nucleotide indels) in at least one of the four *FAD2* genes were identified in approximately 30% and 40% of the shoots regenerated from lines A and B, respectively (Table 1). Shoots with targeted edits in individual and multiple genes, covering all 15 possible LOF genotypes, were identified at different frequencies, with mutations in genotypes 1 through 4 representing the most frequently observed group (Table 1).

To estimate the frequency of unintended mutations at four computationally predicted reagent-mediated off-target loci, a subset of 11 randomly selected regenerated shoots was screened by whole genome sequencing (WGS). Subsequent analysis did not detect any alteration of genomic sequence when compared to untreated shoots at any of the off-target loci examined.

**Table 1.** Distribution of genotypes among regenerated shoots of two canola lines with precise LOF edits ($N_{+1}$, $N_{-1}$, and $N_{-2}$) in four *BnFAD2* genes.

| Number of LOF Genes | LOF Genotype Number | Targeted Genes * | Number of Shoots with Precise Edits | |
|---|---|---|---|---|
| | | | Line A | Line B |
| 1 | 1 | 1 | 130 | 71 |
| | 2 | 2 | 106 | 67 |
| | 3 | 3 | 161 | 174 |
| | 4 | 4 | 118 | 96 |
| 2 | 5 | 1 + 2 | 51 | 6 |
| | 6 | 1 + 3 | 62 | 19 |
| | 7 | 1 + 4 | 34 | 8 |
| | 8 | 2 + 3 | 58 | 21 |
| | 9 | 2 + 4 | 37 | 8 |
| | 10 | 3 + 4 | 121 | 41 |
| 3 | 11 | 1 + 2 + 3 | 29 | 2 |
| | 12 | 1 + 2 + 4 | 12 | 0 |
| | 13 | 1 + 3 + 4 | 91 | 13 |
| | 14 | 2 + 3 + 4 | 68 | 1 |
| 4 | 15 | 1 + 2 + 3 + 4 | 51 | 1 |

* 1, *BnaA05.FAD2a*; 2, *BnaC05.FAD2a*; 3, *BnaA01.FAD2b*; 4, *BnaC05FAD2b*.

From previously published reports [41–45,47], plants with LOF modifications in genes or alleles encoding oleate desaturase (FAD2) have an increased amount of oleic acid (18:1Δ9cis) and decreased amounts of either or both linoleic acid (18:2Δ9,12) and linolenic acid (18:3Δ9,12,15) in the seed oil. The increase in oleic acid content is often concomitant with a decrease in polyunsaturated fatty acids (18:2 and/or 18:3) and will therefore have higher oxidative stability. Several of the gene-edited lines analyzed showed similar patterns of trait variation for oleic acid, linoleic acid, and linolenic acid when compared to wildtype fatty acid levels, demonstrating the efficacy of the targeted gene edits induced in *BnaFAD2* genes to deliver the LOF phenotype (Table 2).

**Table 2.** Seed oil fatty acid composition of *FAD2* gene-edited lines containing different numbers of *FAD2* edits in line A and of a non-edited control grown in the greenhouse. Results for three select fatty acids are shown to demonstrate the impact of the gene edits made in test materials. Data are presented as mean of corresponding biological replications (*n*). The mean differences were analyzed using one-way ANOVA, followed by Fisher's LSD. The differences were measured for significance at $p \leq 0.05$ level. The same letter indicates no significant difference at $p \leq 0.05$ among comparisons in the same column. ANOVA, analysis of variance; LSD, Least Significant Difference.

| Genotype | Number of Edited *FAD2* Genes | Oleic (18:1) | Linoleic (18:2) | Linolenic (18:3) |
|---|---|---|---|---|
| | | Mole% | Mole% | Mole% |
| Wildtype (*n* = 1) | 0 | 61.13 ± 0.00 a | 20.18 ± 0.00 | 12.34 ± 0.00 a |
| *BnaA05.FAD2a; BnaC05.FAD2a (n = 3)* | 2 | 82.73 ± 3.92 b | 3.45 ± 0.79 | 7.35 ± 2.36 b |
| *BnaC05.FAD2a; BnaC01.FAD2b (n = 1)* | 2 | 69.51 ± 0.00 a | 13.64 ± 0.00 | 10.52 ± 0.00 ab |
| *BnaA05.FAD2a; BnaC05.FAD2a; BnaC01.FAD2b(n = 3)* | 3 | 88.42 ± 0.50 c | 1.89 ± 0.05 a | 3.59 ± 0.32 c |
| *BnaA05.FAD2a; BnaC05.FAD2a; BnaA01.FAD2b BnaC01.FAD2b (n = 2)* | 4 | 88.52 ± 0.22 bc | 1.89 ± 0.02 a | 3.47 ± 0.01 c |

Fatty acid levels are expressed in mole% ± SD.

### 3.2. Single Cell Approach—Pod Shatter Reduction

Seed loss due to pod shattering in canola represents a major problem for harvest management in production fields across the main growing region in the northern United States and western Canada, as well as globally. Despite this challenge, there is a general lack of germplasm with native resistance to pod shattering in canola [48,49]. Similar to the

approach taken for the *BnaFAD2* LOF mutant project described herein, we developed novel LOF alleles in the *SHP* gene family, resulting in different genotypic combinations in elite lines of *B. napus* [50]. These lines were propagated over multiple generations and tested in both greenhouse and field growing environments, where the impact of different LOF gene combinations on the pod shatter reduction trait phenotype was assessed.

We co-delivered GRONs BnaSHP-G01, BnaSHP-G02, and BnaSHP-G03 (Table S1), along with CRISPR/Cas9 RNPs BnaSHP-01, BnaSHP-02, BnaSHP-03, and BnaSHP-04, (Table S2) into canola protoplasts using a PEG-mediated delivery approach [17]. GRON-templated mutations of $N_{+1}$, $N_{-1}$, or $N_{-2}$ indels were found in 5–20% of the shoots regenerated from four different canola lines and represented 30–60% of all 255 possible combinations of LOF-targeted genotypes (Table 3). While representing a lower overall frequency, LOF mutations in seven and even eight genes were observed in individual shoots, indicating that multiplex targeting of a large gene family is achievable using this approach.

**Table 3.** Distribution of genotypes among regenerated shoots with precise LOF edits in eight pod shatter reduction (PSR) genes of four canola lines.

| Number of LOF Genes | Number of Targeted Genotypes | Number of Shoots with Precise Edits * | | | |
|---|---|---|---|---|---|
| | | Line A | Line B | Line C | Line D |
| 1 | 8 | 296 | 8 | 145 | 153 |
| 2 | 28 | 234 | 16 | 71 | 87 |
| 3 | 56 | 203 | 23 | 50 | 73 |
| 4 | 70 | 154 | 11 | 15 | 26 |
| 5 | 56 | 114 | 12 | 13 | 25 |
| 6 | 28 | 81 | 10 | 8 | 18 |
| 7 | 8 | 39 | 2 | 10 | 12 |
| 8 | 1 | 7 | 1 | 1 | 2 |
| Number of shoots screened: | | 5395 | 1490 | 2620 | 4620 |

* Precise edits: $N_{+1}$, $N_{-1}$, and $N_{-2}$.

To assess the frequency of unintended mutations at 52 computationally predicted reagent-mediated off-target loci for RNPs BnaSHP-01, BnaSHP-02, BnaSHP-03, and BnaSHP-04, a subset of 71 shoots was screened using targeted amplicon sequencing. Analysis revealed off-target mutations in 16 of the 71 shoots screened, which were primarily restricted to three off-target loci unique to RNP BnaSHP-04 that exhibited three mismatches to the on-target sequence. This relatively high frequency of unintended reagent-mediated mutations is likely related to the initial on-target gRNA design aimed at a family of MADS-box transcription factors. Consequently, relevant alterations in gRNA design have been implemented to mitigate off-target activity at these three off-target loci.

Following the same process as what was used for the aforementioned oil quality gene-edited lines, regenerated shoots bearing different genotype combinations of targeted gene edits which were devoid of reagent-mediated off-target mutations at the 52 computationally predicted loci were transferred to soil and hardened in growth chambers. After a 1–2-week acclimatization period to ex vitro growth conditions, lines were transferred to a controlled greenhouse environment for seed increase and evaluation of the targeted pod shatter reduction trait phenotype. During the initial greenhouse seed increase cycle, regenerated shoots were assessed for overall true-to-type plant morphology. Similar to the observations made on gene-edited lines for the oil quality trait, very few regenerated shoots from the gene editing process displayed an off-type pattern of morphological growth characteristics, with the majority of regenerated plant lines containing target gene edits showing normal plant growth when compared to the isogenic wildtype parent line. Individual regenerated plants were also genotyped to confirm the stable presence of the LOF allele(s) induced by the gene editing process. Genotyping results confirmed that target loci of selected individuals were homozygous for the LOF allele(s). Plants were grown to maturity in the

greenhouse, and pods were collected for evaluation of the pod shatter reduction trait using a random impact test (Figure 2).

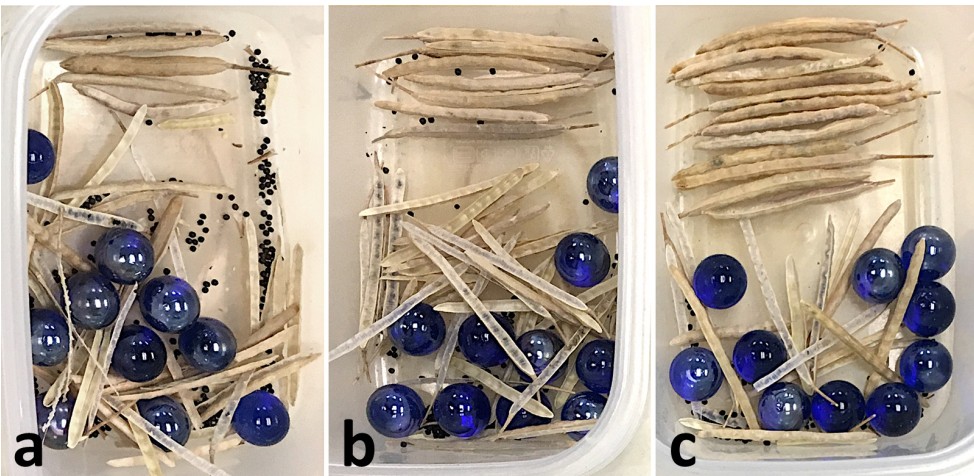

**Figure 2.** Results of a random impact test using marbles on (**a**) wildtype and (**b**,**c**) gene-edited lines containing LOF alleles. Lines shown are considered near-isogenic, with gene-edited lines having been selected for specific genotypes in the target locus from the precision gene editing process. Lines with known gene edits perform more favorably in a random impact test compared to wildtype, as evidenced by visual observation of greater levels of seed dispersal and broken pods in the wildtype tray compared to trays containing gene-edited lines.

Individual seed lines demonstrating the reduction in pod shattering in initial greenhouse tests were advanced for two generations under single seed descent to increase seeds for testing in the field. Gene-edited lines with various LOF mutant genotype combinations were grown in replicated, small plot trials in multiple locations in the upper midwestern US and western Canada. Using the random impact test (RIT), we were able to quantify the impact of the various combinations of mutant LOF alleles derived from our gene editing approach (Table 4). For several of the gene-edited lines tested in both greenhouse and field conditions, a statistically significant difference of pod shatter reduction above the isogenic wildtype line was observed. There was also good correlation ($R^2 = 0.69$) between the greenhouse and field grown response to the RIT across the tested lines.

**Table 4.** Evaluation of additional pod shatter breakage force (BF) required for gene-edited lines in greenhouse and field trial.

| Line | Greenhouse | | Field | |
|---|---|---|---|---|
| | N | % Additional BF | N | % Additional BF |
| A_01 | 3 | 44.44 a | 4 | 48.43 a |
| A_02 | 1 | 42.85 ab | 4 | 51.09 a |
| A_03 | 2 | 35.71 abc | 4 | 37.11 a |
| A_04 | 2 | 30.95 abcd | 4 | 38.80 a |
| A_05 | 3 | 25.55 bcd | 4 | 43.76 a |
| A_06 | 3 | 23.65 cd | 4 | 26.67 a |
| A_07 | 3 | 18.88 cd | 4 | 33.98 a |
| A_08 | 3 | 16.34 de | 4 | 38.63 a |
| A_WT | 3 | 0.00 e | 4 | 0.00 b |
| LSD (0.05) | (3,3) | 17.4 | (4,4) | 25.3 |

Gene-edited lines containing different combinations of loss of function alleles were compared to the wildtype (WT) isogenic background grown in two different environments—Greenhouse (2017) and Field (2019). Data shown represent the percentage of additional breakage force (BF) required above WT isogenic line (Line A) to rupture mature canola pods when tested by a random impact test in the lab. Numbers with different letters in columns are significantly different according to Fisher's LSD ($p \leq 0.05$).

## 4. Discussion

### 4.1. Modified Oil Profiles Using Gene Editing

Modified canola oils are important commercially, principally in the food and specialty chemical markets. The resulting seed oils might have oleic acid contents ranging from 65% to greater than 90% in the form of mixed triglycerides. Vegetable oils with higher oleic acid are generally regarded as desirable due to their stability. This is particularly advantageous for cooking, extended storage, and shelf life. The results of gene-edited products in this study have resulted in canola plants whose seed is at the high end of this desirable range and have shown that we have canola oils with higher oleic acid (C18:1)

### 4.2. Canola with Pods Displaying Pod Shatter Reduction

Using the RIT, Zaman et al. [51] demonstrated that the combination of five mutated Shatterproof homologs in *B. napus* (*BnaA09g55330D*, *BnaC04g23360D*, *BnaA04g01810D*, *BnaC04g52620D*, and *BnaA05g02990D*) generated a significantly higher resistance to pod shattering when compared to the wildtype parent. In these gene-edited lines developed by integrating the CRISPR-Cas9 transgenically, the gRNAs targeted the SHP genes downstream of the MADS-box domain. Similarly, we obtained many pentuple SHP gene-edited lines, as well as hextuple, heptuple, and octuple edited canola lines (Table 3). However, unlike the paper [51], the gRNAs used in this publication targeted the SHP coding sequence upstream of the DNA-binding MADS-box domain.

In the field, it is common practice during canola production for the farmer to swath or windrow their field prior to full senescence of the pods and seeds to avoid pre-harvest seed lost due to pod drying and breakage, resulting in seed dispersal in the field. This is especially so when wind, rain, hail, and other adverse weather conditions occur at maturity, which can further exacerbate seed loss due to pod shattering. Canola lines with a higher resistance to pod breakage are therefore desired in breeding programs for this crop. Despite the need and desire for varieties with a greater level of pod shatter reduction, canola breeders have struggled to consistently identify germplasm with native trait efficacy that can be used to introgress pod shatter reduction characteristics into elite varieties which can be sold commercially. Various mutagenesis methods have thus far been the most successful in consistently delivering commercially acceptable levels of pod shatter reduction to canola [52]. However, the development timelines and potential for collateral damage using random mutagenesis can be inefficient and difficult for breeders to assimilate into their variety development process due to higher costs and the unpredictability associated with the outcome. However, by using a precision gene editing approach, as we have articulated herein, the ability to rapidly assimilate a complex target of mutant alleles into elite lines of canola was demonstrated. The suite of technologies used in IGE represents an improved methodology for accelerating variety development and short-circuiting traditional methods of trait discovery, such as mutagenesis, transgenesis, and conventional breeding, among others. An added benefit of this platform is to develop trait-bearing elite lines with no linkage drag, which would typically occur when traits are developed using other technologies that require backcrossing. This is especially true and valuable for traits with complex genetics, such as pod shatter reduction in canola.

## 5. Conclusions

### 5.1. Industrialized Gene Editing (IGE)

The gene editing and cell biology processes demonstrated in this paper were used at a manufacturing or industrialized scale. IGE is non-transgenic, both in process and product, as it is able to precisely edit plant genes without the integration of foreign genetic material or recombinant DNA. It uses elite germplasm as the starting material for the gene editing process, making trait development and trait combinations more efficient. It enables gene edits to be made in single cells which are regenerated to whole plants possessing the desired traits more quickly and efficiently than other comparable technologies. It is standardized, precise, and reproducible, making trait development customizable and

trait combinations efficient and rapid. It is scalable and has been partially automated to further accelerate the trait development process. It saves time in breeding programs since it eliminates generations of backcrossing to introgress the desired traits into elite parental lines and results in elite lines that have no donor linkage drag.

### 5.2. Multiplex/Multitrait Gene Editing

The concept of IGE in crops will provide another powerful tool in the plant breeder's toolbox. In the past, farmers used selective breeding to obtain desirable phenotypic traits in wild crop plants. Over hundreds of years, farmers have selected for crop variability in this wild *Brassica* plant to achieve its various forms [53]. For example, by selecting and breeding desirable traits such as bigger leaves or larger buds, kale and cauliflower, respectively, were selected through this long and arduous process. More recently, conventional breeding techniques have been used to bring together the variation underlying more complex traits, involving many independent loci, such as yield improvements or disease resistance in commercially relevant crop varieties. Today, in as little as one year, not decades, IGE can develop desirable traits in an elite variety by using the natural processes of the plant. This process can be automated, which not only increases throughput but scales gene editing in plants into a manufacturing process. This will allow IGE to have a huge positive impact on both farmers and consumers in a greatly reduced timeline.

By the year 2050, the United Nations estimates that the human population is expected to increase from 8 billion currently to 9.7 billion. Add the negative effects of climate change on plant growth and we are on the cusp of a crisis of epic proportions. We need to develop characteristics in plants which provide improvements to yields, disease resistance, water-use efficiencies, nutrient-use efficiencies, and drought tolerance, among other beneficial traits, and we need them today. Innovation enables a bridge between sustainability and profit, with these traits enabling farmers to more sustainably steward their land while making every acre more productive. The industrialization of gene editing is part of a solution, and this will allow the global human population and our fragile environment to not only survive in the future but to flourish.

**Supplementary Materials:** The following supporting information can be downloaded at: https://www.mdpi.com/article/10.3390/ijpb14040077/s1, Table S1: Sequence of GRONs used in this study; Table S2: Protospacer sequence of gRNAs used in this study.

**Author Contributions:** J.N.-V., J.M., Z.N., P.L., S.S., C.S. and N.S. conceived the original screening and research plans; A.W., J.N.-V., J.R. and Z.N. supervised the experiments; C.S. provided technical assistance to J.N.-V.; J.N.-V. and J.M. designed the experiments; A.W. and P.L. analyzed the data; G.G. and P.B. conceived the project; N.S. and J.M. wrote the article, with contributions of all the authors; J.M. and N.S. supervised and completed the writing. N.S. agrees to serve as the author responsible for contact and ensures communication. All authors have read and agreed to the published version of the manuscript.

**Funding:** This research received no external funding.

**Institutional Review Board Statement:** Not applicable.

**Informed Consent Statement:** Not applicable.

**Data Availability Statement:** Data are contained within the article or Supplementary Materials.

**Acknowledgments:** We would like to thank all members of the Cibus team, current and past, for their many contributions to the exciting field of precision gene editing technologies in plants. Further, we would like to thank Tony Moran, Amir Sattarzadeh, and Daniel Gobena for critical review of this article.

**Conflicts of Interest:** All authors (except J.N.-V.) are employees of Cibus Inc., an industry leader in developing and applying precision gene editing tools to meet agricultural, industrial, and human health needs. Cibus is a global company with offices in Europe and North America, including its state-of-the-art research and development campus in San Diego, California.

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
