# Peer review of "Industrial Scale Gene Editing in Brassica napus"

_2037-0164, doi:10.3390/ijpb14040077_

Round 1

Reviewer 1 Report

Comments and Suggestions for Authors

This paper analyzes a high-throughput gene editing method, and the results demonstrate certain value. However, the logic of the paper is poor, with different parts mixed together, causing difficulties for readers in reading and understanding, and there is a certain degree of lack of method description.

1. I am not sure if the statement 'Industrialized gene editing' is correct. Carefully consider it. Please provide corresponding references.

2. The abstract needs to clarify the research significance, methods, and results. The current version is almost entirely an introduction to the background, without reflecting the methods and results, and must be rewritten.

3. Most of the content in the introduction reflects conventional operations and does not highlight the characteristics or innovative points of this article. Figure 1 also cannot reflect the difference between what the author refers to as "industrialized gene editing" and conventional operations.

4. If a scientific term appears multiple times in the text, it is recommended to use abbreviated form. For example, 'Industrialized gene editing'.

5. The description of the method should not appear in the result analysis, and the corresponding content needs to be moved into the method section. As shown in lines 234-247.

6. The results and discussion need to be written in two parts.

7. The conclusion is too lengthy and requires significant simplification.

8. Please provide the specific parameters and process of "gas chromatography flame ionization detection (GC FID)", or add references.

9. Table 2 needs to be supplemented with significance analysis.

10. The method related to Figure 2 is not described and needs to be added and the basis or reliability of this method should be explained. In addition, many of results lacks descriptions of corresponding methods, and the author needs to carefully revise and add them.

11. “5. Patents”, is this a necessary part of the paper? I don't think this is a conventional representation method.

Author Response

  1. I am not sure if the statement 'Industrialized gene editing' is correct. Carefully consider it. Please provide corresponding references. Added “We introduce here the concept of industrialized gene editing…” to replace “Industrialized gene editing is …”
  2. The abstract needs to clarify the research significance, methods, and results. The current version is almost entirely an introduction to the background, without reflecting the methods and results, and must be rewritten. Completed. 
  1. Most of the content in the introduction reflects conventional operations and does not highlight the characteristics or innovative points of this article. Figure 1 also cannot reflect the difference between what the author refers to as "industrialized gene editing" and conventional operations. I tried to address this by adding details to the abstract. Figure 1 explains the concept if industrialized gene editing used in this manuscript to develop gene-edited plants for both high oleic acid and pod shatter reduction traits in canola.
  2. If a scientific term appears multiple times in the text, it is recommended to use abbreviated form. For example, 'Industrialized gene editing'. Issue was addressed (Industrialized gene editing (IGE))
  1. The description of the method should not appear in the result analysis, and the corresponding content needs to be moved into the method section. As shown in lines 234-247. Moved to Materials and Methods. Also moved the part where the Random Impact test is described to M&M.
  2. The results and discussion need to be written in two parts. Completed
  3. The conclusion is too lengthy and requires significant simplification. Completed
  4. Please provide the specific parameters and process of "gas chromatography flame ionization detection (GC FID)", or add references. Completed
  5. Table 2 needs to be supplemented with significance analysis. Completed
  6. The method related to Figure 2 is not described and needs to be added and the basis or reliability of this method should be explained. Completed

In addition, many of results lacks descriptions of corresponding methods, and the author needs to carefully revise and add them. Not addressed

  1. “5. Patents”, is this a necessary part of the paper? I don't think this is a conventional representation method. Removed.

Reviewer 2 Report

Comments and Suggestions for Authors

The authors claimed to establish an industrialized, fast and efficient gene editing and screening technology in Brassica napus to facilitate the creation of rapeseed germplasm resources with excellent agronomic traits, and produced precise loss of function edits for two agronomically important complex traits, high oleic oil and pod shatter reduction, in elite canola varieties. However, some issues still need to be addressed properly.

(1)Significant difference analysis should be conducted on the data in Table 2.

(2)There are some formatting errors needed to be corrected, for instance, 1.25 x 105 cells/ml in line 156 should be 1.25 x 105 cells/mL; 90-110 μmol/m2s in line 158 should be 90-110 μmol/m2s; R2 = 0.69 in line 332 should be R2 = 0.69.

(3)All gene names in the manuscript should be italicized.

(4)The Latin name of the species should also be italicized in the whole manuscript.

(5)A brief introduction in the preface needs be added for why choosing FAD2 and SHP for gene editing.

(6)Parameters for “random impact test using marblesand “Determination of seed oil content” should be supplemented in detail in the methods section.

Comments on the Quality of English Language

The authors claimed to establish an industrialized, fast and efficient gene editing and screening technology in Brassica napus to facilitate the creation of rapeseed germplasm resources with excellent agronomic traits, and produced precise loss of function edits for two agronomically important complex traits, high oleic oil and pod shatter reduction, in elite canola varieties. However, some issues still need to be addressed properly.

(1)Significant difference analysis should be conducted on the data in Table 2.

(2)There are some formatting errors needed to be corrected, for instance, 1.25 x 105 cells/ml in line 156 should be 1.25 x 105 cells/mL; 90-110 μmol/m2s in line 158 should be 90-110 μmol/m2s; R2 = 0.69 in line 332 should be R2 = 0.69.

(3)All gene names in the manuscript should be italicized.

(4)The Latin name of the species should also be italicized in the whole manuscript.

(5)A brief introduction in the preface needs be added for why choosing FAD2 and SHP for gene editing.

(6)Parameters for “random impact test using marblesand “Determination of seed oil content” should be supplemented in detail in the methods section.

Author Response

(1)Significant difference analysis should be conducted on the data in Table 2. Completed

(2)There are some formatting errors needed to be corrected, for instance, 1.25 x 105 cells/ml in line 156 should be 1.25 x 105 cells/mL; 90-110 μmol/m2s in line 158 should be 90-110 μmol/m2s; R2 = 0.69 in line 332 should be R2 = 0.69. Corrected

(3)All gene names in the manuscript should be italicized. Corrected

(4)The Latin name of the species should also be italicized in the whole manuscript. Corrected

(5)A brief introduction in the preface needs be added for why choosing FAD2 and SHP for gene editing. Completed.

(6)Parameters for “random impact test using marbles” and “Determination of seed oil content” should. Completed.

Reviewer 3 Report

Comments and Suggestions for Authors

The abstract needs to be revised and the data obtained presented.

A description of the statistical method must be provided in the Materials and Methods section.

 Describe what method was used to determine ploidy.

 The conclusion must be carefully revised, shortened and redone in accordance with your data.

 The list of references must be corrected according to the requirements of the journal.

Author Response

The abstract needs to be revised and the data obtained presented. Corrected

A description of the statistical method must be provided in the Materials and Methods section. Completed.

Describe what method was used to determine ploidy. Corrected

Added (new Line 168) Corrected

The conclusion must be carefully revised, shortened and redone in accordance with your data. Corrected

The list of references must be corrected according to the requirements of the journal. Corrected

Round 2

Reviewer 1 Report

Comments and Suggestions for Authors

The author has made sufficient revisions to the paper. I think it can meet the requirements for publication.